# Exploring the Genomic Landscape of Hepatobiliary Cancers to Establish a Novel Molecular Classification System

**DOI:** 10.3390/cancers16020325

**Published:** 2024-01-11

**Authors:** Anthony J. Scholer, Rebecca K. Marcus, Mary Garland-Kledzik, Debopriya Ghosh, Miquel Ensenyat-Mendez, Joshua Germany, Juan A. Santamaria-Barria, Adam Khader, Javier I. J. Orozco, Melanie Goldfarb

**Affiliations:** 1Division of Surgical Oncology, University of South Carolina School of Medicine, Greenville, SC 29605, USA; joshua.germany@prismahealth.org; 2Department of Surgery, Saint John’s Cancer Institute at Providence St. John’s Health Center, Santa Monica, CA 90404, USA; rebecca.k.marcus@gmail.com (R.K.M.); javier.orozco@providence.org (J.I.J.O.); melanie.goldfarb@providence.org (M.G.); 3Department of Surgery, Division of Surgical Oncology, West Virginia University, Morgantown, WV 26506, USA; mollym.garland@gmail.com; 4Janssen Research and Development LLC, Early Development and Oncology, Biostatistics, Raritan, NJ 08869, USA; debopriya.ghosh@rutgers.edu; 5Cancer Epigenetics Laboratory, Health Research Institute of the Balearic Islands, 07120 Palma, Spain; miki716@gmail.com; 6Department of Surgery, Division of Surgical Oncology, University of Nebraska Medical Center, Omaha, NE 68105, USA; juan.santamaria.barria@gmail.com; 7Department of Surgery, Division of Surgical Oncology, Hunter Holmes McGuire Veterans Affairs Medical Center, Richmond, VA 23249, USA; adammkhader@gmail.com

**Keywords:** next-generation sequencing, hepatobiliary cancer, molecular subtypes, oncogenic pathways

## Abstract

**Simple Summary:**

Machine learning identified hepatobiliary molecular subtypes with overlapping oncogenomic pathways independent of traditional cancer taxonomy. This new classification system identified tumor biology features that may affect future hepatobiliary staging systems and lead to the improved selection of oncologic treatment plans.

**Abstract:**

Taxonomy of hepatobiliary cancer (HBC) categorizes tumors by location or histopathology (tissue of origin, TO). Tumors originating from different TOs can also be grouped by overlapping genomic alterations (GA) into molecular subtypes (MS). The aim of this study was to create novel HBC MSs. Next-generation sequencing (NGS) data from the AACR-GENIE database were used to examine the genomic landscape of HBCs. Machine learning and gene enrichment analysis identified MSs and their oncogenomic pathways. Descriptive statistics were used to compare subtypes and their associations with clinical and molecular variables. Integrative analyses generated three MSs with different oncogenomic pathways independent of TO (*n* = 324; *p* < 0.05). HC-1 “hyper-mutated-proliferative state” MS had rapidly dividing cells susceptible to chemotherapy; HC-2 “adaptive stem cell-cellular senescence” MS had epigenomic alterations to evade immune system and treatment-resistant mechanisms; HC-3 “metabolic-stress pathway” MS had metabolic alterations. The discovery of HBC MSs is the initial step in cancer taxonomy evolution and the incorporation of genomic profiling into the TNM system. The goal is the development of a precision oncology machine learning algorithm to guide treatment planning and improve HBC outcomes. Future studies should validate findings of this study, incorporate clinical outcomes, and compare the MS classification to the AJCC 8th staging system.

## 1. Introduction

Hepatobiliary cancers, encompassing intrahepatic, hilar, and distal cholangiocarcinoma (CCA), hepatocellular carcinoma (HCC), and gallbladder cancer (GBC), are characterized by their rarity and typically dire prognosis. This is largely attributable to advanced-stage presentation and the limited efficacy of current treatments [1,2,3,4,5,6,7,8,9,10,11]. Traditional classification of hepatobiliary cancer (HBC) clusters tumors based on anatomic location or histopathology—referred to as the tissue of origin (TO)—and includes HCC, CCA, and GBC. Several classification systems exist, employing these criteria for diagnosis and prognostic stratification [12,13,14,15,16,17]. Notably, the current American Joint Committee on Cancer (AJCC) TNM system also stratifies HBC prognosis by TO [18].

Recent advancements in molecular, genomic, epigenomic, and transcriptomic research signal a potential paradigm shift in HBC taxonomy and staging systems [19,20]. There is a growing body of evidence suggesting that tumors from differing TOs can be grouped based on similar genomic alterations (GAs), leading to a molecular classification or subtyping [21,22,23,24,25,26,27,28]. This approach is exemplified by the latest AJCC breast cancer staging, which has adopted a Prognostic Stage (PS) [29,30,31]. The development of the PS addressed limitations of prior staging systems that failed to adequately discern between breast cancer subtypes based on tumor biology and response to treatment [32,33,34,35]. This modern staging system integrates molecular subtypes, such as hormone and HER-2 receptor status, with genomic profiles, and a 21-gene assay Recurrence Score (RS), to more accurately predict treatment outcomes [36].

In light of these developments, we contend that there is a critical need for an updated HBC taxonomy and staging systems that integrate molecular classifications. We hypothesize that the tumor biology, prognosis, and treatment response of HBC are significantly influenced by their molecular subtypes’ oncogenomic pathways. Therefore, this proof-of-concept study was conducted with the objective to identify molecular subtypes (MS) unique to HBC, with the intent to enhance our understanding of HBC tumor biology.

## 2. Materials and Methods

### 2.1. Selection of a Genomic Database 

We conducted a multidimensional analysis of next-generation sequencing (NGS) data to unravel the genomic landscape of hepatobiliary cancers (HBCs), utilizing the American Association of Cancer Research (AACR) Genomics Evidence Neoplasia Information Exchange (GENIE) Project. The AACR-GENIE, an international precision oncology collaborative, compiles genomic data linked with electronic medical records from 19 leading cancer centers, providing an expansive registry [37,38,39]. Protocols for data generation and sharing of de-identified information are institutionally approved across all participating entities of AACR-GENIE.

### 2.2. Gene Panels Used for Next-Generation Sequencing of Hepatobiliary Cancers

The gene panels from the AACR-GENIE database (version 6.1) employed in this study are detailed in the Pipeline for Annotating Mutations and Filtering Putative Germline Single Nucleotide Polymorphisms (SNPs), available at https://www.aacr.org/wp-content/uploads/2020/02/20200127_GENIE_Data_Guide_7.pdf (accessed on 2 March 2023) [37,38,39]. This compendium included 48 genes consistently represented across the gene panels used by contributing institutions. Data encompass mutation profiles, copy-number variants from three centers, and structural rearrangements from two centers. Paired tumor-normal sequencing was conducted at two centers, while the remaining executed tumor-only sequencing [37,38,39]. To enhance the validity of our findings, we excluded patients with incomplete gene mutation coverage based on the specific genomic panels used.

### 2.3. Patient Selection 

From the AACR-GENIE database, 329 patients aged 18–89 years with hepatobiliary cancer were identified and classified into three TO subtypes: cholangiocarcinoma (CCA, *n* = 115), hepatocellular carcinoma (HCC, *n* = 127), and gallbladder cancer (GBC, *n* = 87) (Figure 1). The CCA TO subtype included intrahepatic, perihilar, and distal cholangiocarcinomas due to the database’s present classification system. Exclusion criteria entailed patients with non-hepatobiliary cancers, hepatoblastoma, liver angiosarcoma, co-diagnosis of HCC and intrahepatic cholangiocarcinoma (ICC), or limited genomic sequencing data (Figure 1). Predominantly, a single tissue sample was subjected to NGS, with primary tumor data utilized for patients with dual samples.

### 2.4. Processing of Genomic Data 

The AACR-GENIE database (version 6.1) was retrieved via the cBioPortal and Synapse Platform (https://genie.cbioportal.org, accessed on 2 March 2023) using R Version 6.1 and R Studio Version 1.3.1073 (The R Foundation, Vienna, Austria). Compiled genomic data included CNVs, translocations, SNVs, INDELs, single-point variants, and gene fusions [37,38,39]. Data were aggregated into a dichotomous variable denoting the presence or absence of pathologic genomic alterations, annotated with marker type, sample barcode, and histopathology subtype [37,38,39]. Analyses focused on alterations listed in the Catalogue of Somatic Mutations in Cancer (COSMIC) Census (http://cancer.sanger.ac.uk/census/, accessed on 2 March 2023) and observed in at least five patients (>1.5%) [40].

### 2.5. Identification of HBC Molecular Subtypes 

A comprehensive evaluation of the molecular data for gene mutations and CNVs was undertaken using machine learning. The initial dataset contained 981 features, including 36 duplicates and 194 CNV events. After excluding empty features and patients lacking CNV data, 786 features across 324 patients remained. *PREX2* was excluded due to inadequate sequencing data, leaving 47 genes. An AI-based deep integrative analysis utilized both mutational and CNV data to identify unique molecular subtypes. This analysis distilled a singular value of molecular alterations with 99 features in 324 patients (excluding 5 for insufficient CNV data); three features with correlated events were removed. Non-negative Matrix Factorization (NMF) machine learning discerned the optimal molecular subtype number, employing the top 38 most variable features. The Uncorrelated Shrunken Centroids (USC) method, set to a delta value below 5 (Δ < 5) and a maximum correlation between genomic regions of 0.7, was used to classify hepatobiliary tumors into distinct molecular subtypes.

### 2.6. Bioinformatics and Biostatistical Analyses

The ‘oncoprint’ feature of the R/Bioconductor package ‘TCGAbiolinks’ visualized the mutational profiles of each HBC molecular subtype. Correlation and unsupervised hierarchical clustering analyses were performed with the MultiExperiment Viewer v4.9 [41,42,43,44]. Gene enrichment pathway analyses utilized Metascape [45] to interpret the MS genomic data. Pearson’s χ2 test analyzed the associations between TO and MS subtypes in 311 patients (excluding 13 for lack of data, Figure 1). Statistical analyses employed the dplyr and tidyr R packages, considering *p* < 0.05 as significant. Summary statistics detailed the cohort and individual subtypes, with Pearson’s χ2 test elucidating associations between categorical variables like TO subtype, molecular subtype, age groups, and race. The R/ggplot2 package facilitated data visualization.

## 3. Results

### 3.1. Genomic Landscape of HBCs Reveals Variation between TO Subtypes

Our comprehensive genomic analysis encompassed 329 HBC patients, predominantly male (57.1%, *n* = 188), white (74.5%, *n* = 245), and non-Hispanic (92.7%, *n* = 305), with a median age of 64 years. Hepatocellular carcinoma (HCC) was the most frequently observed TO subtype (38.6%, *n* = 127), followed by cholangiocarcinoma (CCA) at 35.0% (*n* = 115) and gallbladder cancer (GBC) at 26.4% (*n* = 87). Across these patients, 42 genes manifested a total of 1017 genomic alterations, with 77.8% of tumors presenting at least two alterations. GBCs exhibited a higher mean number of alterations compared to HCC and CCA, yet HCC recorded the highest number of maximum alterations (*p* < 0.05). The most frequently altered genes were *TP53, TERT, CTNNB1, KRAS, ARID1A, CDKN2A, IDH1, PIK3CA, MYC,* and *SMDA4* (*p* < 0.001) (Table 1). *IDH1* and *KRAS* mutations correlated with CCA, *CDKN2A* mutations with CCA and GBC, and *CTNNB1* and *TERT* mutations with HCC (*p* < 0.05). *TP53* and *ARID1A* mutations were present across all TO subtypes (Appendix A). 

### 3.2. Machine Learning Distinguishes Three Unique HBC Molecular Subtypes

Out of the initial cohort, 324 hepatobiliary tumors were subjected to a machine learning algorithm, resulting in the identification of three distinct HBC molecular subtypes, denoted as HC-1, HC-2, and HC-3 (*p* < 0.05) (Figure 2 and Figure 3). Subtype HC-1 presented a hyper-mutated phenotype with prevalent mutations in *TP53* and *ARID1A*, and CNVs in *CCND1* and *FGF4*. This subtype exhibited high CNVs (>5%) across all genes analyzed. HC-2 was marked by pathogenic mutations in *TP53, TERT,* and *CTNNB1*, with generally low CNVs, except in genes like *MYC, RECQL4,* and *CDKN2A*. The HC-3 subtype, characterized by lower mutation rates, displayed significant mutations in *IDH1* and higher CNVs in *MDM2* and *CDKN2A*.

### 3.3. Gene Enrichment Analyses Elucidates Oncogenomic Pathways 

Gene mutation and CNV profiles for each molecular subtype (*n* = 324) guided our gene enrichment analyses (Figure 4). The HC-1 subtype was enriched for alterations in cell growth, mitotic transition, and chromosome organization, indicative of a proliferative phenotype. In contrast, HC-2 tumors were disrupted in pathways akin to stem cell function and cellular senescence, while HC-3 tumors exhibited changes in metabolic processes and stress response pathways (*p* < 0.05).

### 3.4. Comparative Analysis of HBC Molecular and TO Subtypes

Following the establishment of MSs, 311 HBC cases remained for comparative analysis, mirroring the demographics of our initial genomic evaluation: primarily male (57.6%), Caucasian (74.9%), non-Hispanic (92.9%), aged 55–69 years (51.8%), with a median age of 64 years. HCC constituted the largest TO subtype (38.3%), with CCA (34.4%) and GBC (27.3%) following. Gender disparities were noted, with a higher likelihood of HCC in males and GBC in females (*p* < 0.001). No significant differences in TO subtypes were observed in relation to age or ethnicity (Table 2).

The HC-2 molecular subtype was most prevalent, representing 56.9% of HBC tumors and showing a higher frequency in HCC and GBC (Table 2 and Figure 5). Conversely, the HC-3 subtype, comprising 32.5% of the cohort, was predominantly associated with CCA. The least common, HC-1, accounted for 10.6% and was evenly distributed across all TOs. Significant differences in genomic alterations led to distinct mutation profiles for each MS (*p* < 0.001) (Appendix A), with oncogenic mutations being particularly abundant in HC-2 and HCC subtypes (*p* = 0.0005) (Figure 6).

## 4. Discussion

Our genomic analysis of hepatocellular carcinoma (HCC), cholangiocarcinoma (CCA), and gallbladder cancer (GBC) revealed several gene mutations consistent with prior literature [11,22,41,46,47,48,49,50,51,52,53,54,55]. Validating the HBC genomic data within the GENIE database was an essential preliminary step, setting the stage to explore “overlapping” molecular subtypes. The identification of these HBC molecular subtypes (MSs) holds significance for multiple reasons. Notably, despite HBC TO subtypes having a significant proportion of clinically actionable mutations among GI cancers, a vast majority of HBC tumors lack these mutations, leading to poorer outcomes and limited clinical trial inclusion [52,53,54,55,56]. Additionally, the molecular signatures specific to TO subtypes contribute to resistance to therapies [57,58,59,60]. The rarity of HBCs compounds the difficulty of studying these cancers and understanding their complex oncogenomic pathways.

In this study, machine learning facilitated the creation of a modern taxonomy system comprising three HBC-specific MSs. Unlike previous GI malignancy classifications, which included a heterogeneous mix of tumors, the HBC MSs we identified are more homogenized, a feature we believe is critical for unraveling the obscure oncogenomic pathways that influence HBC tumor biology.

Gene enrichment analysis shed light on several oncogenomic pathways associated with each HBC MS, revealing their untapped potential. Our data suggest that HBC patients within the HC-1 subtype, characterized by a “hyper-mutated-proliferative state”, could potentially derive the most benefit from chemotherapy. This subgroup’s predisposition to rapid cell division renders them more susceptible to cytotoxic chemotherapy than slower-dividing cells. This may clarify why previous studies have reported a lackluster response to chemotherapy, given that only a subset of hepatobiliary tumors falls under the HC-1 classification [2,3,4,5,6,61,62,63,64]. Future research should aim to determine the predictive value of the HC-1 classification for chemotherapy responsiveness.

The HC-2 subtype, dubbed the “adaptive stem cell-cellular senescence” subtype, exhibited epigenomic alterations that may enable evasion of immune surveillance and contribute to treatment resistance. These cells can become dormant under treatment-induced stress, potentially resurging when conditions are favorable, which may serve as a mechanism for treatment resistance. Additionally, the prevalence of CTNNB1 mutations in HC-2 tumors suggests a disrupted WNT/beta-catenin pathway. While inhibiting upstream regulators of this pathway has been explored, direct targeting of beta-catenin might improve response in tumors with such mutations [60,65]. The absence of FDA-approved drugs targeting beta-catenin highlights the necessity for research into HC-2 specific therapies to enhance treatment efficacy and survival outcomes for HBC.

The HC-3 “metabolic-stress pathway” subtype, despite its lower mutational load compared to HC-1 and HC-2, features clinically actionable IDH1 mutations. Nevertheless, the majority of HC-3 tumors lacked this mutation, underscoring the limited number of HBC patients suitable for targeted therapy. Alterations in this subtype are implicated in proteomic mutation pathways and metabolism of lipids, fats, and amino acids [48,66]. Further investigation is warranted to elucidate how these metabolic genomic alterations influence tumor biology and to potentially discover novel treatments [67].

An intriguing aspect of this study was the observed disparity in baseline patient characteristics, hinting at potential inequalities in NGS utilization, referrals to high-volume cancer centers, or access to national cancer centers for non-Caucasian HBC patients. Although NGS is covered by Medicare since 2018, recent studies have indicated disparities in NGS usage among different racial and insurance groups in non-HBC malignancies [68]. This finding, not previously confirmed in HBC patients, warrants further research to uncover the underlying causes of such disparities in NGS utilization.

Our study, conducted through retrospective analysis of the ACCR-GENIE database, inherently faces several limitations that warrant a careful interpretation of the results. These limitations include selection bias, confounding factors, and the potential for incomplete data, which could influence the generalizability of our findings. Acknowledging these challenges, we emphasize the need for diverse patient populations and the use of real-world data in future research to enhance the reliability of outcomes and extend their relevance to broader patient groups.

Despite the breadth of the ACCR-GENIE database, the omission of transcriptomic data presents a notable gap in our exploration of oncogenomic pathways. Additionally, the current amalgamation of intrahepatic and extrahepatic cholangiocarcinoma cases under a single category limits our ability to distinguish the nuances between these subtypes. It is anticipated that subsequent updates to the database will enable more refined classification, facilitating deeper insights into the molecular subtypes identified.

The scarcity of detailed clinical data also hampers our ability to draw robust correlations between molecular subtypes and clinical endpoints such as prognosis and treatment response. As the database evolves, incorporating more comprehensive clinical details will be crucial for refining these molecular subtypes and understanding their implications in a clinical context.

Our study sets the stage for future research, aiming to validate and build upon the initial hypothesis. Through subsequent, more inclusive and methodologically robust studies, including phase III clinical trials and multivariable analyses, we aim to mitigate the impact of biases and enhance the external validity of our findings. The ultimate goal is to foster a deeper understanding of hepatobiliary cancer biology and to potentially guide clinical decision-making.

## 5. Conclusions

This study introduces HBC molecular subtypes as a pioneering step toward revolutionizing cancer taxonomy and integrating genomic profiling into HBC staging systems. Our goal is to develop a precision oncology algorithm, informed by machine learning, to refine treatment planning and enhance patient outcomes in HBC. Future studies should validate our findings, incorporate clinical data, and evaluate how our HBC MS classification compares with the current AJCC HBC staging system.

## Figures and Tables

**Figure 1 cancers-16-00325-f001:**
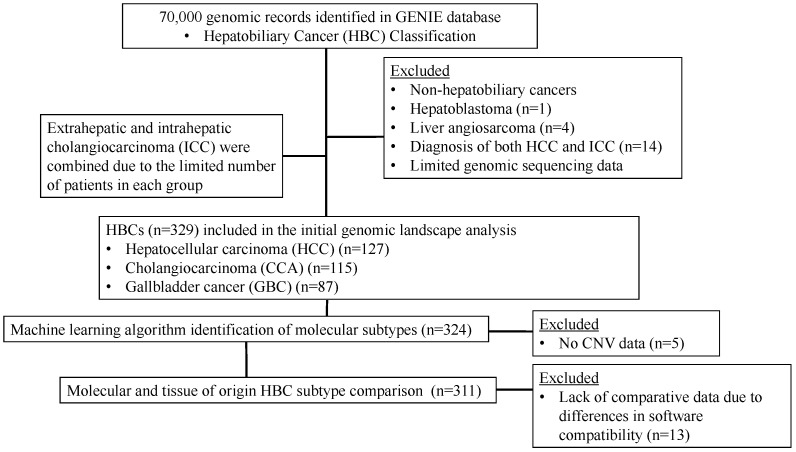
Consortium diagram for hepatobiliary cancer (HBC) patients included in analyses.

**Figure 2 cancers-16-00325-f002:**
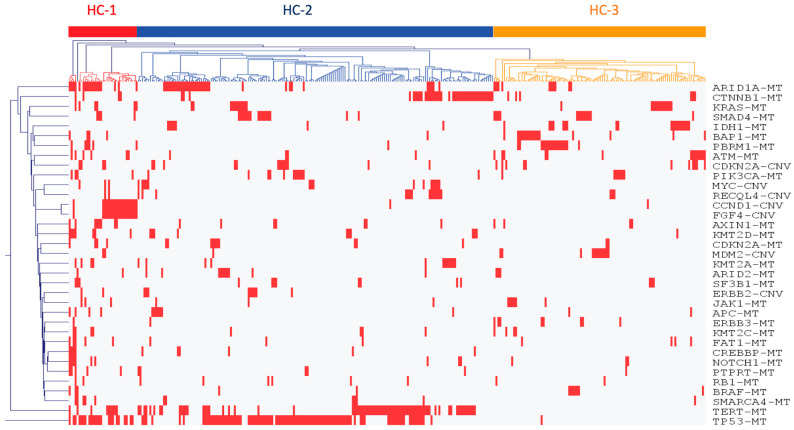
Machine learning identified three novel hepatobiliary cancer molecular subtypes (HC) with distinct gene mutations; for simplicity, these were designated HC-1, HC-2, and HC-3.

**Figure 3 cancers-16-00325-f003:**
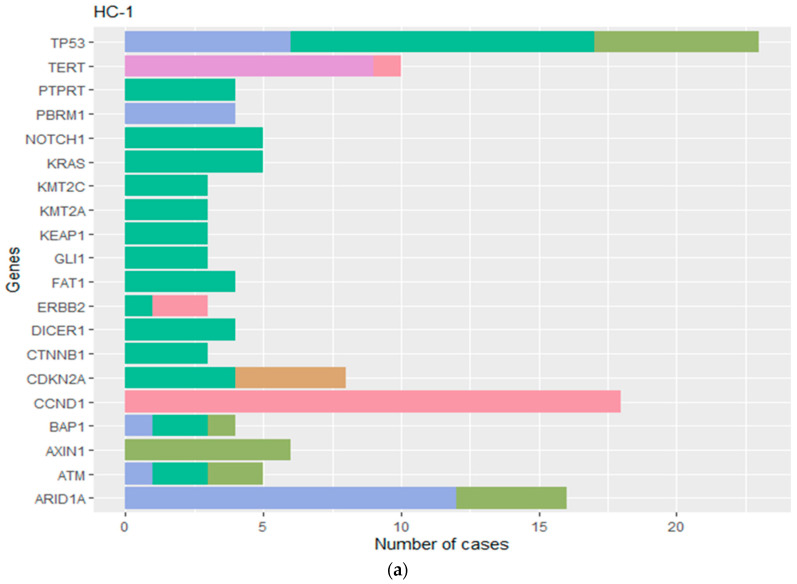
(**a**) The gene alterations of the HC-1 subtype. HC-1 subtype favored gene mutations in *TP53* and *ARIDIA*. (**b**) The gene alterations of the HC-2 subtype. The HC-2 subtype demonstrated gene mutations in *TP53*, *TERT*, and *CTNNB1*. (**c**) The gene alterations of the HC-3 subtype. The HC-3 subtype had mutations in *IDH1*. HC (hepatobiliary cancer molecular subtype).

**Figure 4 cancers-16-00325-f004:**
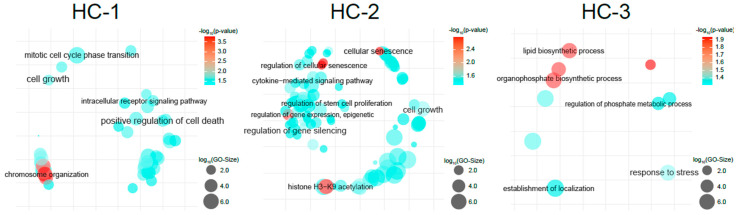
Gene enrichment pathway analyses using molecular subtype genomic data identified different oncogenomic pathways influencing the tumor biology of the distinct molecular subtypes (*p* < 0.05). HC (hepatobiliary cancer molecular subtype).

**Figure 5 cancers-16-00325-f005:**
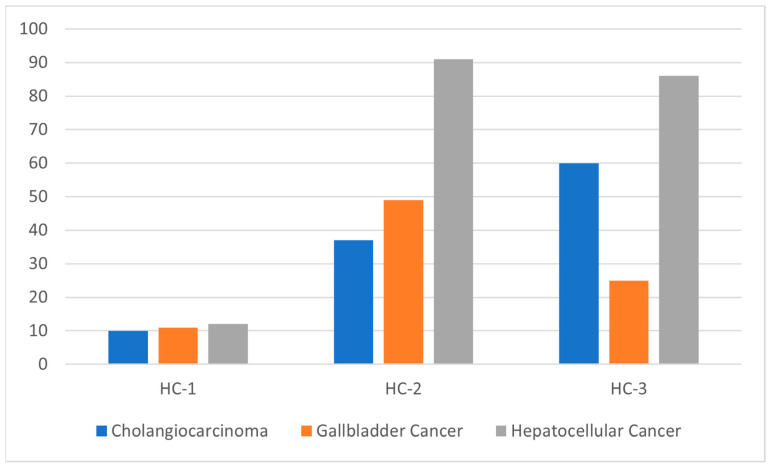
Distribution of tissue of origin subtypes (CCA, GBC, HCC) in molecular subtypes (HC-1, HC-2, HC-3). HC (hepatobiliary cancer molecular subtype), GBC (gallbladder cancer), HCC (hepatocellular cancer), CCA (cholangiocarcinoma).

**Figure 6 cancers-16-00325-f006:**
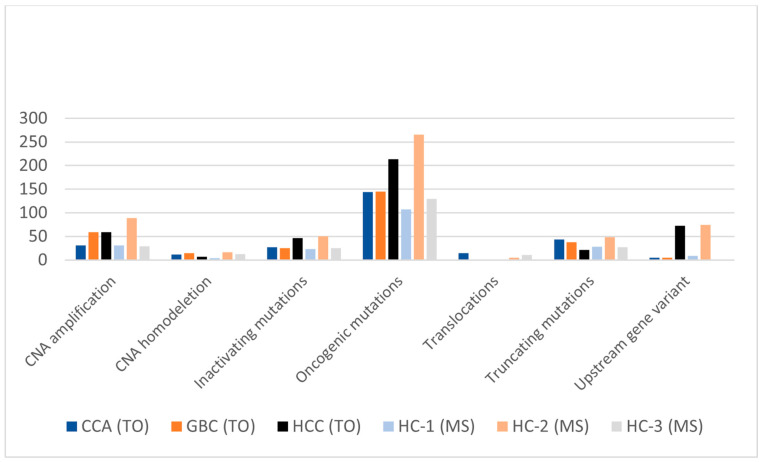
Genomic alterations in hepatobiliary cancer tissue of origin and molecular subtypes. HC (hepatobiliary cancer molecular subtype), GBC (gallbladder cancer), HCC (hepatocellular cancer), CCA (cholangiocarcinoma).

**Table 1 cancers-16-00325-t001:** Gene mutations and genomic alterations in cholangiocarcinoma (CCA), gallbladder cancer (GBC) and hepatocellular carcinoma (HCC) *.

Genes	Alteration Types	CCA	GBC	HCC	Total	X^2^	*p* Value
*TERT*	Upstream gene variant	8	5	90	103	135.51	<0.001
*CTNNB1*	Oncogenic mutations	3	7	71	81	107.85	<0.001
*KRAS*	Oncogenic mutations	50	12	1	63	62.95	<0.001
*IDH1*	Oncogenic mutations	46	1	2	49	80.86	<0.001
*FGFR2*	Translocations	13	1	0	14	22.43	<0.001
*PBRM1*	Truncating mutations	11	2	0	13	15.85	<0.001
*BAP1*	Truncating mutations	10	1	1	12	13.5	0.001
*MDM2*	CNA amplifications	8	13	0	21	12.29	0.002
*CCNE1*	CNA amplifications	3	12	1	16	12.88	0.002
*NFE2L2*	Oncogenic mutations	1	0	8	9	12.67	0.002
*SMAD4*	Inactivating mutations	5	11	0	16	11.38	0.003
*AXIN1*	Inactivating mutations	0	5	11	16	11.38	0.003
*CDKN2A*	CNA homo deletion	24	23	7	54	10.11	0.006
*ATM*	Inactivating mutations	8	0	2	10	10.4	0.006
*JAK1*	Oncogenic mutations	1	1	8	10	9.8	0.007
*PBRM1*	Inactivating mutations	10	3	1	14	9.57	0.008
*KDR*	Oncogenic mutations	4	0	9	13	9.38	0.009
*NRAS*	Oncogenic mutations	9	2	1	12	9.5	0.009
*ERBB2*	CNA amplifications	7	10	0	17	9.29	0.01
*MYC*	CNA amplifications	6	7	18	31	8.58	0.014
*BRAF*	Oncogenic mutations	11	1	6	18	8.33	0.016
*SF3B1*	Oncogenic mutations	11	1	7	19	8	0.018
*ERBB2*	Oncogenic mutations	8	8	0	16	8	0.018
*SMAD4*	Oncogenic mutations	11	13	2	26	7.92	0.019
*SMAD4*	Truncating mutations	3	7	0	10	7.4	0.025
*RAD21*	CNA amplifications	0	3	7	10	7.4	0.025
*IDH2*	Oncogenic mutations	6	0	2	8	7	0.03
*CDK12*	CNA amplifications	2	6	0	8	7	0.03
*ERBB3*	CNA amplifications	2	6	0	8	7	0.03
*KRAS*	CNA amplifications	2	6	0	8	7	0.03
*FGFR2*	Oncogenic mutations	5	0	1	6	7	0.03
*TET1*	Oncogenic mutations	5	1	0	6	7	0.03
*TSC1*	Oncogenic mutations	5	0	1	6	7	0.03
*CDK6*	CNA amplifications	1	5	0	6	7	0.03
*SUZ12*	Oncogenic mutations	0	1	5	6	7	0.03
*FAT1*	Oncogenic mutations	13	4	5	22	6.64	0.036
*PIK3CA*	Oncogenic mutations	16	16	5	37	6.54	0.038
*TP53*	Truncating mutations	7	17	7	31	6.45	0.04
*NTRK1*	CNA amplifications	2	1	7	10	6.2	0.045
*RB1*	Inactivating mutations	1	2	7	10	6.2	0.045

* Analysis performed prior to the creation of molecular subtypes.

**Table 2 cancers-16-00325-t002:** Comparison of hepatobiliary tumor molecular and tissue of origin subtypes.

Overall (*n* = 311)	Molecular Subtypes (*n* = 311)				Tissue of Origin Subtypes (*n* = 311)			
	HC-1 33 (10.6)	HC-2 177 (56.9)	HC-3 101 (32.5)	*p*-Value	CCA 107 (34.4)	GBC 85 (27.3)	HCC 119 (38.3)	*p*-Value
Characteristic								
Sex				0.05				<0.001
Male	18 (10.0)	112 (62.6)	49 (27.4)		61 (34.1)	26 (14.5)	92 (51.4)	
Female	15 (11.4)	65 (49.2)	52 (39.4)		46 (34.8)	59 (44.7)	27 (20.5)	
Age Group				0.22				0.4
<55 years old	7 (9.8)	35 (49.4)	29 (40.8)		30 (42.3)	20 (28.2)	21 (29.5)	
≥55–69 years old	17 (10.6)	90 (55.9)	54 (33.5)		53 (32.9)	45 (28.0)	63 (39.1)	
>69 years old	9 (11.4)	52 (65.8)	18 (22.8)		24 (30.4)	20 (25.3)	35 (44.3)	
Race				0.77				0.01
White	22 (9.4)	132 (56.7)	79 (33.9)		92 (39.5)	60 (25.8)	81 (34.8)	
Asian	4 (13.3)	15 (50.0)	11 (36.7)		5 (16.7)	10 (33.3)	15 (50.0)	
Black	3 (14.3)	12 (57.1)	6 (28.6)		4 (19.1)	10 (47.6)	7 (33.3)	
Other	4 (14.8)	18 (66.7)	5 (18.5)		6 (22.2)	5 (18.5)	16 (59.2)	
Ethnicity				0.57				0.77
Non-Hispanic	30 (10.4)	167 (57.8)	92 (31.8)		98 (33.9)	79 (27.3)	112 (38.8)	
Hispanic	2 (11.8)	9 (52.9)	6 (35.3)		8 (47.1)	4 (23.5)	5 (29.4)	
Unknown	1 (20.0)	1 (20.0)	3 (60.0)		1 (20.0)	2 (40.0)	2 (40.0)	

HC (hepatobiliary cancer molecular subtype), GBC (gallbladder cancer), HCC (hepatocellular cancer), CCA (cholangiocarcinoma). For all results, unless indicated otherwise, data are presented as number (percent) (*n*, (%)).

## Data Availability

The data presented in this study are available upon request from the corresponding author.

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
