# Peer review of "Exploring the Genomic Landscape of Hepatobiliary Cancers to Establish a Novel Molecular Classification System"

_cancers, 2024, doi:10.3390/cancers16020325_

Round 1
Reviewer 1 Report
Comments and Suggestions for Authors
Specific comments to the authors
The authors Anthony J. Scholer et al. of the submitted manuscript “Exploring the Genomic Landscape of Hepatobiliary Cancers to Establish a Novel Molecular Classification System” try to identify new molecular subtypes (MS) of hepatobiliary cancers (HBC) using machine learning and gene enrichment analysis of next-generation sequencing (NGS) data from the AACR-GENIE database.
In summary, based on their in-silico-investigations authors could identify the following three different oncogenomic pathways: (i) a “hyper-34 mutated-proliferative state” MS dividing cells susceptible to chemotherapy; (ii) a “adaptive stem cell-cellular senescence” MS evading immune system and showing treatment-resistant mechanisms; (iii) a “metabolic-stress pathway” demonstrating metabolic alterations. The authors postulated that such MSs could lead to a cancer taxonomy evolution, which consecutively will be incorporated as genomic profiling into the current TNM system.
Overall, the submitted manuscript presents an interesting approach to the classification of hepatobiliary cancers by in silico bioinformatics analysis. The manuscript (including presentation) is comprehensible and convincing. The methods are mostly well described. Although the results and discussion are clearly presented, the authors (see specific comments) must perform some minor to major changes to improve significantly the impact of the manuscript. In conclusion, the presented data are interesting. After incorporating the mentioned specific comments (see below) the manuscript has the potency to be accepted.
Overall comments:
This study raises more questions than it answers. A real validation of the proposed new molecular classification is completely missing, which reduces the significance and impact of the presented study. Furthermore, existing molecular analyses of embryogenesis and human malignancies show clear differences between HBCs subtypes such as intrahepatic, hilar and distal cholangiocarcinoma (CCA), hepatocellular carcinoma (HCC) and gallbladder cancer (GBC), raising the question of whether such an approach of including all HBCs is even possible.
Specific comments
Title: The title does not indicate the in-silico-characteristics of the presented study. Please change adequately.
Results:
# Table 1 and 2: The two tables could be combined, since table 1 does not give more information. The authors should show the distribution of TOs in the MSs, which is missing until now, since it will be very interesting, where the TOs will be distributed in the MSs.
# Figure 1: The combining the data of ICC and ECC is critically since the known molecular data of ICC and ECC revealed significant differences.
# Figure 2: Please add a table in combination to figure 1 showing the significant alterated genes to each MSs, too.
# Figure 3 and 4: Please identify the significant differences in detail. The two figures could be combined to show the differences between MSs and TOs.
Discussion: The authors related their findings to references "43, 48 to 51", which is not comparable as these studies "only" analyse intrahepatic cholangiocarcinoma (43, 48 to 50) or liver carcinoma (50), whereas the authors of the present study include all anatomical regions of possible HBC. Regarding the mentioned references, the author has to compare their three different oncogenomic pathways with known pathways by means of an additional Sankey diagram to show concordances and discrepancies.
Comments on the Quality of English LanguageMinor editing of English language required.
Author Response
REVIEWER 1:
Overall comments:
Reviewer 1 #1. This study raises more questions than it answers. A real validation of the proposed new molecular classification is completely missing, which reduces the significance and impact of the presented study. Furthermore, existing molecular analyses of embryogenesis and human malignancies show clear differences between HBCs subtypes such as intrahepatic, hilar and distal cholangiocarcinoma (CCA), hepatocellular carcinoma (HCC) and gallbladder cancer (GBC), raising the question of whether such an approach of including all HBCs is even possible.
We are thankful for your critical analysis, which correctly points out the exploratory nature of our study. Our research indeed raises new questions, serving as an impetus for further inquiry. This proof-of-concept study was designed to identify molecular subtypes within hepatobiliary cancers that differ from previously characterized, more heterogeneous gastrointestinal MSs that include HBCs. In the process of achieving this aim, we also delineated the molecular pathways associated with each subtype. However, it's crucial to note that our study did not set out to determine the clinical significance, prognosis, or predictive ability of these subtypes or their molecular pathways. The lack of necessary data precluded such an analysis; had this information been available, we certainly would have endeavored to investigate these aspects further to enhance the robustness of our findings.
We acknowledge that validation of the proposed classification is beyond the scope of this initial study.
We concur with your observation regarding the significance of subtype-specific molecular differences in HBCs. While current analyses do present distinct molecular features among intrahepatic, hilar, and distal cholangiocarcinoma (CCA), as well as hepatocellular carcinoma (HCC) and gallbladder cancer (GBC), our approach aims to explore the possibility of overarching molecular subtypes that could have clinical relevance across these traditional subdivisions. Our study posits that a broader classification could potentially inform treatment decisions in the future, after rigorous validation and correlation with clinical outcomes.
The GENIE database's lack of comprehensive clinical data at the time of our study is a limitation that we have transparently acknowledged in our discussion. The database did not include detailed information on tumor characteristics treatment regimens, or overall and progression-free survival, which are crucial for evaluating the prognostic and therapeutic implications of the identified MSs. We aim to address this gap by pursuing collaborative research projects that will integrate clinical data to prospectively validate the MSs and enhance our understanding of their clinical utility.
The ability to stratify HBC beyond histological subtypes holds promise for personalizing systemic therapies. Understanding the molecular variances at a granular level is a prerequisite to achieving this goal. If published, our manuscript would offer a novel perspective on the classification of HBC, advocating for an expanded taxonomy that goes beyond tissue of origin subtypes and their associated targetable mutations.
The inclusion of all HBCs in determining MSs is a challenging but necessary endeavor to address the rarity and heterogeneity of these cancers. As more comprehensive data becomes available, we anticipate that our approach will stimulate robust discussions and catalyze further research to refine these classifications and their clinical correlations.
We appreciate your insights and the opportunity to enhance our manuscript in response to your feedback.
Specific comments:
Reviewer 1 #2. Title: The title does not indicate the in-silico-characteristics of the presented study. Please change adequately.
We value the reviewer’s suggestion regarding the title. However, we intentionally chose a broader title to appeal to a diverse readership, including those with varying levels of genomics knowledge. We believe this choice will foster a wider interest and accessibility to the insights presented in our study.
Results:
Reviewer 1 #3. Table 1 and 2: The two tables could be combined, since table 1 does not give more information. The authors should show the distribution of TOs in the MSs, which is missing until now, since it will be very interesting, where the TOs will be distributed in the MSs.
Upon reviewing the reviewer’s suggestion, we agree that Table 1 offered redundant information compared to Table 2 and have subsequently removed it. We direct the reader’s attention to Figure 5, which provides a detailed distribution of tissue of origin subtypes (TOs) within the molecular subtypes (MSs).
Reviewer 1 #4. Figure 1: The combining the data of ICC and ECC is critically since the known molecular data of ICC and ECC revealed significant differences.
Recognizing the significance of distinct molecular profiles between ICC and ECC, as well as their potential impact on our study's conclusions, we agree that a more detailed categorization would have been ideal. We anticipate that future updates to the GENIE database will allow for this level of detailed classification, enabling subsequent studies to conduct a more focused analysis of cholangiocarcinoma subtypes. Such advancements will be crucial in refining our understanding of the molecular landscape of these cancers and their implications for personalized treatment strategies.
Reviewer 1 #5. Figure 2: Please add a table in combination to figure 1 showing the significant altered genes to each MSs, too.
Thank you for your suggestion regarding Figure 2. We acknowledge the importance of presenting the significant altered genes associated with each molecular subtype to provide a comprehensive view of our findings. Figure 1, designed as a Consortium diagram, indeed did not encompass these details.
In response to your valuable input, we have now included additional figures — specifically, Figure 2 and Figure 3 — in the manuscript to explicitly demonstrate the significant altered genes for each MS. These figures were initially part of our supplemental material, primarily due to constraints on the number of figures allowed in the main manuscript. However, recognizing the critical nature of this information for a complete understanding of the molecular subtypes, we decided to incorporate them into the main document.
The inclusion of these figures aims to enhance the depth and clarity of our data presentation, allowing readers to more effectively grasp the genomic distinctions between the molecular subtypes we have identified in hepatobiliary cancers. We believe that this addition will significantly contribute to the comprehensiveness and scientific value of our study.
Reviewer 1 #6. Figure 3 and 4: Please identify the significant differences in detail. The two figures could be combined to show the differences between MSs and TOs.
We thank you for your constructive suggestion regarding Figures 3 and 4. Following your advice, we have merged these figures into a single, comprehensive Figure 6. This updated figure is intentionally designed to clearly demonstrate the significant differences in genomic alterations across the various hepatobiliary cancer subtypes.
It is important to clarify that the p-value of 0.0005, as indicated in the merged figure, specifically pertains to the differences in types of genomic alterations within individual HBC subtypes. However, it does not reflect differences between molecular subtypes and tissue of origin subtypes, owing to the current limitations in our capacity for statistical analysis which restrict our ability to perform a detailed mathematical comparison between MSs and TOs.
We hope that Figure 6, with its integrated and detailed presentation, effectively fulfills your request and enhances the clarity of our findings. This figure is intended to facilitate a more nuanced understanding of the genomic landscape across HBC subtypes, which is crucial for the progression of precision oncology.
Discussion:
Reviewer 1 #7. The authors related their findings to references "43, 48 to 51", which is not comparable as these studies "only" analyse intrahepatic cholangiocarcinoma (43, 48 to 50) or liver carcinoma (50), whereas the authors of the present study include all anatomical regions of possible HBC.
We thank you for bringing to our attention the concern regarding the comparability of our study to the referenced literature. We have taken your feedback into consideration and revised our manuscript to include a broader spectrum of references that encompass HCC, CCA, and GBC genomic studies. These additional references (11, 21, 22, 24, 52-54) have been carefully selected to provide a more aligned comparison, ensuring that they collectively represent a homogenized view of the genomic landscape across the spectrum of hepatobiliary cancers.
Furthermore, our analysis has identified several gene mutations consistent with the findings from these studies, which serves to validate the genomic data of HBC within the GENIE database and reinforces the foundation of our research on molecular subtypes. This meticulous approach to referencing allows us to construct a well-founded platform that acknowledges the complexity and diversity inherent in HBCs and supports the investigation of overlapping molecular subtypes. We have made sure to reflect these updates in the manuscript for a clearer, more accurate context.
We believe these changes address your concerns and enhance the manuscript's scientific rigor by ensuring that comparisons are made with studies that are methodologically and contextually relevant to our work.
Reviewer 1 #8: Regarding the mentioned references, the author has to compare their three different oncogenomic pathways with known pathways by means of an additional Sankey diagram to show concordances and discrepancies.
We sincerely value your suggestion to employ a Sankey diagram for illustrating the oncogenomic pathways of the hepatobiliary cancer molecular subtypes discovered in our study. The importance of such visual representations in elucidating the concordances and discrepancies between our findings and established pathways is well understood.
The rarity of HBCs indeed poses significant challenges in studying these cancers and achieving a comprehensive understanding of their oncogenomic pathways. The majority of studies tend to aggregate HBCs with other gastrointestinal malignancies, with only a few focusing on individual tissue of origin subtypes. Our study strives to fill this gap by identifying overlapping molecular subtypes that encompass HCC, CCA, and GBC TOs. This approach is crucial as it underpins our hypothesis that specific MS oncogenomic pathways can profoundly affect tumor biology, prognosis, and treatment resistance in HBC.
We acknowledge the potential value a Sankey diagram could provide in comparing the MS oncogenomic pathways identified in our study with those documented in the literature. However, given that the molecular subtypes we have reported may represent novel entities not previously characterized, a direct comparison is inherently challenging. This is compounded by the fact that we lack the resources necessary to create a Sankey diagram at this juncture.
Nevertheless, we have endeavored to update our manuscript with comprehensive tables and figures that succinctly highlight the most common gene mutations and oncogenic pathways of the molecular subtypes identified. These additions serve to offer an organized and visually informative presentation of our findings, which we believe effectively communicates the novel insights of our research in lieu of a Sankey diagram.
We are committed to advancing the understanding of HBC and appreciate your engagement, which significantly contributes to the quality and impact of our research.

Reviewer 2 Report
Comments and Suggestions for Authors
Comments to the authors:
In this study, the authors explored the Genomic Landscape of Hepatobiliary Cancers to Establish a Novel Molecular Classification System. I have some concerns that need to be addressed by the authors before publication. The specific comments see below:
1: Clarity and Structure: The text lacks a clear introduction that outlines the problem statement, objectives, and significance of the study. A brief overview at the beginning could enhance reader comprehension. The transition between sections could be smoother. Consider providing clearer headings and subheadings to guide the reader through the content.
2: Data Presentation: The data presentation is dense, and the information could be better organized using figures or tables. Visual aids can help readers grasp complex genomic data more efficiently. The use of terms such as "HC-1," "HC-2," and "HC-3" is introduced without prior explanation, making it challenging for readers to understand their significance. Definitions or explanations should be provided when introducing such terminology.
3: Validation and Generalization: The text emphasizes the need for future studies to validate the findings. However, it lacks discussion on the limitations of the current study, and potential sources of bias or confounding factors are not addressed. The generalizability of the findings to a broader population or other datasets should be acknowledged. Any specific population characteristics that may limit generalization should be discussed.
4: Integration with Existing Frameworks: The text mentions the incorporation of genomic profiling into the TNM system but does not provide sufficient detail on how this integration might occur. Further discussion on the implications for clinical practice and existing cancer classification systems is needed.
5: Clinical Relevance: While the study identifies molecular subtypes, there is limited discussion on the clinical implications of these subtypes. How might these findings impact treatment strategies or patient outcomes? This aspect needs to be elaborated.
6: Language and Terminology: The text uses technical terms without sufficient explanation, making it challenging for readers without a strong background in genomics. Strive for clarity and consider adding a glossary or footnotes for complex terms.
7: References: If applicable, include recent references that support the study's foundation, especially in rapidly evolving fields like genomics. The following paper should be cited: CI 2023, 2 (1), 16; https://doi.org/10.58567/ci02010002
Addressing these points will help enhance the overall clarity, validity, and applicability of the study.
Comments on the Quality of English Language
Minor editing of English language required
Author Response
REVIEWER 2:
Comments to the authors:
In this study, the authors explored the Genomic Landscape of Hepatobiliary Cancers to Establish a Novel Molecular Classification System. I have some concerns that need to be addressed by the authors before publication. The specific comments see below:
REVIEWER 2 #1. Clarity and Structure: The text lacks a clear introduction that outlines the problem statement, objectives, and significance of the study. A brief overview at the beginning could enhance reader comprehension. The transition between sections could be smoother. Consider providing clearer headings and subheadings to guide the reader through the content.
We acknowledge the importance of clarity and structure in presenting our research, particularly in the introduction where establishing a clear narrative is crucial for reader comprehension. We have revised the manuscript to include a succinct overview at the beginning, which now clearly outlines the problem statement, objectives, and significance of our study on hepatobiliary cancer (HBC) molecular subtypes.
To improve the transition between sections and enhance the overall flow of the manuscript, we have refined our headings and subheadings. These revisions aim to guide the reader more intuitively through our content, from the introduction of the research topic to the presentation of our findings and conclusions.
These changes will provide a coherent structure to the manuscript, making it more accessible to readers and effectively conveying the critical aspects of our research. We appreciate your feedback, which has been instrumental in improving the manuscript's presentation and organization.
REVIEWER 2 #2. The data presentation is dense, and the information could be better organized using figures or tables. Visual aids can help readers grasp complex genomic data more efficiently.
We are thankful for your constructive critique regarding the presentation of our data. In response to your observations, we have incorporated additional tables and figures (Figures 3-6, Table 1), as outlined in our responses to previous comments. These enhancements have been carefully designed to alleviate the density of the information and provide a more structured and accessible visual representation of the complex genomic data.
We understand that the intricate nature of genomic studies can be challenging to navigate, and we believe that these new visual aids will significantly aid in the reader's comprehension and engagement with the data. We trust that these improvements will facilitate a more intuitive understanding of our findings and convey the nuances of our research more effectively.
We are committed to clear and informative scientific communication and hope that these modifications will address your concerns satisfactorily.
REVIEWER 2 #3. The use of terms such as "HC-1," "HC-2," and "HC-3" is introduced without prior explanation, making it challenging for readers to understand their significance. Definitions or explanations should be provided when introducing such terminology.
We thank you for your astute observation regarding the introduction of terminology in our manuscript. Your point is well taken; clarity in scientific communication is paramount, especially when introducing novel concepts and classifications.
In the updated manuscript, we have ensured that each instance of the terms "HC-1," "HC-2," and "HC-3" is accompanied by a clear definition upon their first mention, to acquaint readers with their significance immediately. These terms represent the three unique molecular subtypes of hepatobiliary cancer identified in our study through AI-based deep integrative analysis.
Additionally, gene enrichment analyses for each molecular subtype have been elaborated upon, offering insights into the distinct oncogenomic pathways that define each subtype's unique characteristics.
To ensure the reader's ease of understanding, these definitions and accompanying explanations are provided at the initial point of discussion within Sections 3.2 and 3.3 of the manuscript, as well as being clearly annotated within the relevant tables and figures. Lastly, a glossary of terms as been added to the manuscript.
Validation and Generalization:
REVIEWER 2 #4. The text emphasizes the need for future studies to validate the findings. However, it lacks discussion on the limitations of the current study, and potential sources of bias or confounding factors are not addressed.
We appreciate your observation regarding the discussion of limitations and potential biases in our study. In the revised manuscript, we have addressed these concerns comprehensively in the final paragraph of the discussion section.
This study, by its nature as a retrospective analysis utilizing a database, is subject to several inherent limitations. These include, but are not limited to, selection bias, confounding factors, and the accuracy and completeness of the reported data. We recognize that these issues can influence how the results are interpreted and their applicability to broader patient populations.
To mitigate these limitations in future research, we will seek to incorporate diverse patient populations, employ real-world data, and engage in phase III clinical studies. The utilization of multivariable analyses will further help to minimize the impact of known and unknown confounding factors and biases, thereby enhancing the reliability of our findings.
We also acknowledge the limitations arising from the ACCR-GENIE database, such as the lack of transcriptomic data and the aggregation of intrahepatic and extrahepatic cholangiocarcinoma into a single category. These factors restrict our ability to fully explore the oncogenomic pathways and the nuanced behaviors of different subtypes since, if we had access to transcriptomic data, we could analyze if the mutated pathways are also altered at a transcriptomic level. We anticipate that future iterations of the database will allow for a more detailed classification, which will be invaluable for differentiating these subtypes and elucidating their characteristics in relation to the molecular subtypes we have identified.
Moreover, the paucity of detailed clinical data in the database at the time of our study has limited our capacity to predict prognosis and treatment response based on the molecular subtypes. The lack of survival data is especially important, as we cannot analyze if some subtypes are more aggressive than others. As more comprehensive data become available, we expect to refine these subtypes and better understand their clinical implications.
The aim of our proof-of-concept study was to identify HBC-specific MSs and to establish a baseline for future hypothesis-driven research. The findings from this study are intended to stimulate new lines of investigation, leading to a more profound understanding of HBC biology and potentially informing clinical practice in the future.
REVIEWER 2 #6. The generalizability of the findings to a broader population or other datasets should be acknowledged. Any specific population characteristics that may limit generalization should be discussed.
We thank you for your comment on the generalizability of our findings. We fully agree that extending the applicability of our results to a broader patient population or other datasets is a crucial next step in our research.
In the updated manuscript, we have addressed the need for future studies that will aim to validate our findings across diverse patient cohorts. While the GENIE database does include patients from a variety of hospitals, which provides some level of heterogeneity, we acknowledge that this does not suffice for the level of external validation required for the widespread clinical application of molecular subtypes.
To that end, we have underscored the necessity of conducting studies with broader and more varied patient populations. These studies should ideally be designed to account for demographic and geographic variations that may affect the presentation and progression of hepatobiliary cancers.
Moreover, we recognize that incorporating our MS into clinical practice must be preceded by rigorous, externally validated research. Such studies will help determine whether the molecular subtypes we have identified hold significant prognostic or predictive value when applied in real-world clinical settings.
REVIEWER 2 #7. Integration with Existing Frameworks: The text mentions the incorporation of genomic profiling into the TNM system but does not provide sufficient detail on how this integration might occur. Further discussion on the implications for clinical practice and existing cancer classification systems is needed.
We acknowledge your insightful request for a more detailed discussion on how genomic profiling could be integrated with the TNM staging system and its implications for clinical practice. In our proof-of-concept study, which utilized the limited clinical data available in the ACCR-GENIE database, we have indeed referenced the potential for such integration but did not delve into the specifics, given the early stage of our research.
Looking forward, we aim to conduct studies that will correlate the HBC molecular subtypes (MSs) with clinical outcomes. This is a complex process that would likely require the maturation of clinical data over time to fully understand the prognostic and predictive utility of these MSs. In our endeavor to integrate genomic profiling into the TNM system, we anticipate drawing on insights from breast cancer research, where hormone receptor status, HER-2 expression, and genomic assays have been incorporated to refine prognostic staging.
The evolution of cancer taxonomy is indeed progressing, with an increasing number of studies demonstrating the feasibility of clustering tumors based on genomic alterations, regardless of their tissue of origin. For instance, the American Joint Committee on Cancer's most recent staging system for breast cancer has adopted a Prognostic Stage that includes molecular subtype information, which has improved the predictive accuracy for treatment response and outcomes.
Our manuscript's introduction and relevant sections have been updated to reflect these points and to articulate the potential pathway towards integrating HBC MSs into the existing TNM staging system. We envision that this will evolve similarly to the advancements seen in breast cancer, where molecular subtyping has significantly contributed to personalized treatment strategies and improved patient outcomes.
REVIEWER 2 #8. Clinical Relevance: While the study identifies molecular subtypes, there is limited discussion on the clinical implications of these subtypes. How might these findings impact treatment strategies or patient outcomes? This aspect needs to be elaborated.
We appreciate your request for additional detail regarding the potential integration of our molecular subtypes with existing cancer staging frameworks such as the TNM system.
Our study's primary aim was to identify unique molecular subtypes within hepatobiliary cancers (HBCs), distinct from the more heterogeneous subtypes previously described in gastrointestinal malignancies. While we have successfully highlighted the existence of these HBC-specific subtypes, it is important to note that our work did not extend to the analysis of the clinical utility of these subtypes. Without the necessary clinical data, we cannot ascertain the full implications of our findings for prognosis, treatment response, or survival outcomes.
This proof-of-concept study has opened avenues for further research, which we hope will include the application of these molecular subtypes to a prospective validation cohort. As outlined in our response to Reviewer 1, this subsequent research would aim to determine the practical utility of these subtypes in clinical settings.
In anticipation of future studies, we acknowledge the need for a detailed plan to integrate genomic profiling into the TNM system. We envision a multi-phase process that begins with validating our molecular subtypes against clinical outcomes and eventually establishing their role in enhancing the prognostic accuracy of the TNM system.
By laying this initial groundwork, we aim to provide a platform for future studies to build upon, potentially leading to a more nuanced and precise approach to cancer classification that could directly impact clinical decision-making and patient care.
REVIEWER 2 #9.Language and Terminology: The text uses technical terms without sufficient explanation, making it challenging for readers without a strong background in genomics. Strive for clarity and consider adding a glossary or footnotes for complex terms.
We thank you for the feedback on the use of technical terminology in our manuscript. In our endeavor to make our research accessible to a wider audience, including those who may not have an extensive background in genomics, we have included a glossary of terms. This glossary provides clear definitions and explanations for specialized terms used throughout the text.
We understand the importance of clear and precise communication in scientific discourse, especially when the subject matter involves complex genomic concepts. Our intention is to facilitate the reader's understanding and ensure that the significance of our findings is conveyed effectively to all members of the scientific community, irrespective of their specialty.
We trust that the inclusion of this glossary will enhance the manuscript's clarity and serve as a valuable resource for readers, aiding in the comprehension of the genomic data and interpretations presented in our study.
REVIEWER 2 #10. References: If applicable, include recent references that support the study's foundation, especially in rapidly evolving fields like genomics. The following paper should be cited: CI 2023, 2 (1), 16; https://doi.org/10.58567/ci02010002
We are grateful for your recommendation to include additional recent references pertinent to our study's domain. Following your guidance, we attempted to locate the specified document using the provided DOI: 10.58567/ci02010002. Unfortunately, our efforts to retrieve the paper via PubMed were not successful, as the reference did not yield any results nor provided a title that could guide us to the pertinent study.
We fully acknowledge the importance of integrating the latest research to substantiate our study, particularly in a field as dynamic as genomics. If you could provide further details regarding the title, authors, or the context in which this reference should be included, we would be more than willing to review and incorporate this citation into our manuscript.
We understand the value of a well-supported manuscript that reflects current scientific advances and are committed to ensuring that our references are comprehensive and up-to-date. We appreciate your assistance in enhancing the quality of our citations and await any additional information you can provide regarding the recommended reference.
